# Distributed Submodular Cover:
# Succinctly Summarizing Massive Data

**Baharan Mirzasoleiman**
ETH Zurich

**Amin Karbasi**
Yale University

**Ashwinkumar Badanidiyuru**
Google

**Andreas Krause**
ETH Zurich

## Abstract

How can one find a subset, ideally as small as possible, that well represents a massive dataset? I.e., its corresponding utility, measured according to a suitable utility function, should be comparable to that of the whole dataset. In this paper, we formalize this challenge as a submodular cover problem. Here, the utility is assumed to exhibit submodularity, a natural diminishing returns condition prevalent in many data summarization applications. The classical greedy algorithm is known to provide solutions with logarithmic approximation guarantees compared to the optimum solution. However, this sequential, centralized approach is impractical for truly large-scale problems. In this work, we develop the first distributed algorithm – DISCOVER – for submodular set cover that is easily implementable using MapReduce-style computations. We theoretically analyze our approach, and present approximation guarantees for the solutions returned by DISCOVER. We also study a natural trade-off between the communication cost and the number of rounds required to obtain such a solution. In our extensive experiments, we demonstrate the effectiveness of our approach on several applications, including active set selection, exemplar based clustering, and vertex cover on tens of millions of data points using Spark.

## 1 Introduction

A central challenge in machine learning is to extract useful information from massive data. Concretely, we are often interested in selecting a small subset of data points such that they maximize a particular quality criterion. For example, in nonparametric learning, we often seek to select a small subset of points along with associated basis functions that well approximate the hypothesis space [1]. More abstractly, in data summarization problems, we often seek a small subset of images [2], news articles [3], scientific papers [4], etc., that are representative w.r.t. an entire corpus. In many such applications, the utility function that measures the quality of the selected data points satisfies submodularity, i.e., adding an element from the dataset helps more in the context of few selected elements than if we have already selected many elements (c.f., [5]).

Our focus in this paper is to find a succinct summary of the data, i.e., a subset, ideally as small as possible, which achieves a desired (large) fraction of the utility provided by the full dataset. Hereby, utility is measured according to an appropriate submodular function. We formalize this problem as a submodular cover problem, and seek efficient algorithms for solving it in face of massive data. The celebrated result of Wolsey [6] shows that a greedy approach that selects elements sequentially in order to maximize the gain over the items selected so far, yields a logarithmic factor approximation. It is also known that improving upon this approximation ratio is hard under natural complexity theoretic assumptions [7]. Even though such a greedy algorithm produces near-optimal solutions,

it is impractical for massive datasets, as sequential procedures that require centralized access to the full data are highly constrained in terms of speed and memory.

In this paper, we develop the first distributed algorithm – DISCOVER – for solving the submodular cover problem. It can be easily implemented in MapReduce-style parallel computation models [8] and provides a solution that is competitive with the (impractical) centralized solution. We also study a natural trade-off between the communication cost (for each round of MapReduce) and the number of rounds. The trade-off lets us choose between a small communication cost between machines while having more rounds to perform or a large communication cost with the benefit of running fewer rounds. Our experimental results demonstrate the effectiveness of our approach on a variety of submodular cover instances: vertex cover, exemplar-based clustering, and active set selection in non-parametric learning. We also implemented DISCOVER on Spark [9] and approximately solved vertex cover on a social graph containing more than $65$ million nodes and $1.8$ billion edges.

## 2  Background and Related Work

Recently, submodular optimization has attracted a lot of interest in machine learning and data mining where it has been applied to a variety of problems including viral marketing [10], information gathering [11], and active learning [12], to name a few. Like convexity in continuous optimization, submodularity allows many discrete problems to become efficiently approximable (e.g., constrained submodular maximization).

In the submodular cover problem, the main objective is to find the smallest subset of data points such that its utility reaches a desirable fraction of the entire dataset. As stated earlier, the sequential, centralized greedy method fails to appropriately scale. Once faced with massive data, MapReduce [8] (and modern implementations like Spark [9]) offer arguably one of the most successful programming models for reliable parallel computing. Distributed solutions for some special cases of the submodular cover problem have been recently proposed. In particular, for the set cover problem (i.e., find the smallest subcollection of sets that covers all the data points), Berger et al. [13] provided the first distributed solution with an approximation guarantee similar to that of the greedy procedure. Blelloch et al. [14] improved their result in terms of the number of rounds required by a MapReduce-based implementation. Very recently, Stergiou et al. [15] introduced an efficient distributed algorithm for set cover instances of massive size. Another variant of the set cover problem that has received some attention is maximum $k$-cover (i.e., cover as many elements as possible from the ground set by choosing at most $k$ subsets) for which Chierichetti et al. [16] introduced a distributed solution with a $(1 - 1/e - \epsilon)$ approximation guarantee.

Going beyond the special case of coverage functions, distributed constrained submodular maximization has also been the subject of recent research in the machine learning and data mining communities. In particular, Mirzasoleiman et al. [17] provided a simple two-round distributed algorithm called GREEDI for submodular maximization under cardinality constraints. Contemporarily, Kumar et al [18] developed a multi-round algorithm for submodular maximzation subject to cardinality and matroid constraints. There have also been very recent efforts to either make use of randomization methods or treat data in a streaming fashion [19, 20]. To the best of our knowledge, we are the first to address the general distributed submodular cover problem and propose an algorithm DISCOVER for approximately solving it.

## 3  The Distributed Submodular Cover Problem

The goal of data summarization is to select a small subset $A$ out of a large dataset indexed by $V$ (called the ground set) such that $A$ achieves a certain quality. To this end, we first need to define a utility function $f : 2^V \to \mathbb{R}_+$ that *measures* the quality of any subset $A \subseteq V$, i.e., $f(A)$ quantifies how well $A$ represents $V$ according to some objective. In many data summarization applications, the utility function $f$ satisfies *submodularity*, stating that the gain in utility of an element $e$ in context of a summary $A$ decreases as $A$ grows. Formally, $f$ is submodular if

$$f(A \cup \{e\}) - f(A) \geq f(B \cup \{e\}) - f(B),$$

for any $A \subseteq B \subseteq V$ and $e \in V \setminus B$. Note that the meaning of utility is application specific and submodular functions provide a wide range of possibilities to define appropriate utility functions. In

Section 3.2 we discuss concrete instances of functions $f$ that we consider in our experiments. Let us denote the marginal utility of an element $e$ w.r.t. a subset $A$ as $\triangle(e|A) = f(A \cup \{e\}) - f(A)$. The utility function $f$ is called *monotone* if $\triangle(e|A) \geq 0$ for any $e \in V \setminus A$ and $A \subseteq V$. Throughout this paper we assume that the utility function is monotone submodular.

The focus of this paper is on the *submodular cover problem*, i.e., finding the smallest set $A^c$ such that it achieves a utility $Q = (1 - \epsilon)f(V)$ for some $0 \leq \epsilon \leq 1$. More precisely,

$$A^c = \arg\min_{A \subseteq V} |A|, \quad \text{such that} \quad f(A) \geq Q. \tag{1}$$

We call $A^c$ the optimum *centralized* solution with size $k = |A^c|$. Unfortunately, finding $A^c$ is NP-hard, for many classes of submodular functions [7]. However, a simple greedy algorithm is known to be very effective. This greedy algorithm starts with the empty set $A_0$, and at each iteration $i$, it chooses an element $e \in V$ that maximizes $\triangle(e|A_{i-1})$, i.e., $A_i = A_{i-1} \cup \{\arg\max_{e \in V} \triangle_f(e|A_{i-1})\}$. Let us denote this (centralized) greedy solution by $A^g$. When $f$ is *integral* (i.e., $f : 2^V \to \mathbb{N}$) it is known that the size of the solution returned by the greedy algorithm $|A^g|$ is at most $H(\max_e f(\{e\}))|A^c|$, where $H(z)$ is the $z$-th harmonic number and is bounded by $H(z) \leq 1 + \ln z$ [6]. Thus, we have $|A^g| \leq (1 + \ln(\max_e f(\{e\})))|A^c|$, and obtaining a better solution is hard under natural complexity theoretic assumptions [7]. As it is standard practice, for our theoretical analysis to hold, we assume that $f$ is an integral, monotone submodular function.

**Scaling up: Distributed computation in MapReduce.** In many data summarization applications where the ground set $V$ is large, the sequential greedy algorithm is impractical: either the data cannot be stored on a single computer or the centralized solution is too expensive in terms of computation time. Instead, we seek an algorithm for solving the submodular cover problem in a distributed manner, preferably amenable to MapReduce implementations. In this model, at a high level, the data is first distributed to $m$ machines in a cluster, then each part is processed by the corresponding machine (in parallel, without communication), and finally the outputs are either merged or used for the next round of MapReduce computation. While in principle multiple rounds of computation can be realized, in practice, expensive synchronization is required after each round. Hence, we are interested in distributed algorithms that require few rounds of computation.

## 3.1 Naive Approaches Towards Distributed Submodular Cover

One way of solving the distributed submodular cover problem in multiple rounds is as follows. In each round, all machines – in parallel – compute the marginal gains for the data points assigned to them. Then, they communicate their best candidate to a central processor, who then identifies the globally best element, and sends it back to all the $m$ machines. This element is then taken into account when selecting the next element with highest marginal gain, and so on. Unfortunately, this approach requires synchronization after each round and we have exactly $|A^g|$ many rounds. In many applications, $k$ and hence $|A^g|$ is quite large, which renders this approach impractical for MapReduce style computations.

An alternative approach would be for each machine $i$ to select greedily enough elements from its partition $V_i$ until it reaches at least $Q/m$ utility. Then, all machines merge their solution. This approach is much more communication efficient, and can be easily implemented, e.g., using a single MapReduce round. Unfortunately, many machines may select redundant elements, and the merged solution may suffer from diminishing returns and never reach $Q$. Instead of aiming for $Q/m$, one could aim for a larger fraction, but it is not clear how to select this target value.

In Section 4, we introduce our solution DISCOVER, which requires few rounds of communication, while at the same time yielding a solution competitive with the centralized one. Before that, let us briefly discuss the specific utility functions that we use in our experiments (described in Section 5).

## 3.2 Example Applications of the Distributed Submodular Cover Problem

In this part, we briefly discuss three concrete utility functions that have been extensively used in previous work for finding a diverse subset of data points and ultimately leading to good data summaries [1, 17, 21, 22, 23].

**Truncated Vertex Cover:** Let $G = (V, E)$ be a graph with the vertex set $V$ and edge set $E$. Let $\varrho(C)$ denote the neighbours of $C \subseteq V$ in the graph $G$. One way to measure the influence of a set $C$

is to look at its cover $f(C) = |\varrho(C) \cup C|$. It is easy to see that $f$ is a monotone submodular function. The truncated vertex cover is the problem of choosing a small subset of nodes $C$ such that it covers a desired fraction of $|V|$ [21].

**Active Set Selection in Kernel Machines:** In many application such as feature selections [22], determinantal point processes [24], and GP regression [23], where the data is described in terms of a kernel matrix $K$, we want to select a small subset of elements while maintaining a certain diversity. Very often, the utility function boils down to $f(S) = \log \det(I + \alpha K_{S,S})$ where $\alpha > 0$ and $K_{S,S}$ is the principal sub-matrix of $K$ indexed by $S$. It is known that $f$ is monotone submodular [5].

**Exemplar-Based Clustering:** Another natural application is to select a small number of exemplars from the data representing the clusters present in it. A natural utility function (see, [1] and [17]) is $f(S) = L(\{e_0\}) - L(S \cup \{e_0\})$ where $L(S) = \frac{1}{|V|} \sum_{e \in V} \min_{v \in S} d(e, v)$ is the $k$-medoid loss function and $e_0$ is an appropriately chosen reference element. The utility function $f$ is monotone submodular [1]. The goal of distributed submodular cover here is to select the smallest set of exemplars that satisfies a specified bound on the loss.

## 4 The DISCOVER Algorithm for Distributed Submodular Cover

On a high level, our main approach is to reduce the submodular cover to a sequence of cardinality constrained submodular maximization problems[1], a problem for which good distributed algorithms (e.g., GREEDI [17, 25, 26]) are known. Concretely, our reduction is based on a combination of the following three ideas.

To get an intuition, we will first assume that we have access to an optimum algorithm which can solve cardinality constrained submodular maximization exactly, i.e., solve, for some specified $\ell$,

$$A^{\mathrm{oc}}[\ell] = \arg \max_{|S| \leq \ell} f(S). \tag{2}$$

We will then consider how to solve the problem when, instead of $A^{\mathrm{oc}}[\ell]$, we only have access to an approximation algorithm for cardinality constrained maximization. Lastly, we will illustrate how we can parametrize our algorithm to trade-off the number of rounds of the distributed algorithm versus communication cost per round.

### 4.1 Estimating Size of the Optimal Solution

Momentarily, assume that we have access to an optimum algorithm OPTCARD$(V, \ell)$ for computing $A^{\mathrm{oc}}[\ell]$ on the ground set $V$. Then one simple way to solve the submodular cover problem would be to incrementally check for each $\ell = \{1, 2, 3, \ldots\}$ if $f(A^{\mathrm{oc}}[\ell]) \geq Q$. But this is very inefficient since it will take $k = |A^c|$ rounds of running the distributed algorithm for computing $A^{\mathrm{oc}}[\ell]$. A simple fix that we will follow is to instead start with $\ell = 1$ and double it until we find an $\ell$ such that $f(A^{\mathrm{oc}}[\ell]) \geq Q$. This way we are guaranteed to find a solution of size at most $2k$ in at most $\lceil \log_2(k) \rceil$ rounds of running $A^{\mathrm{oc}}[\ell]$. The pseudocode is given in Algorithm 1. However, in practice, we cannot run Algorithm 1. In particular, there is no efficient way to identify the optimum subset $A^{\mathrm{oc}}[\ell]$ in set $V$, unless P=NP. Hence, we need to rely on approximation algorithms.

### 4.2 Handling Approximation Algorithms for Submodular Maximization

Assume that there is a distributed algorithm DISCARD$(V, m, \ell)$, for cardinality constrained submodular maximization, that runs on the dataset $V$ with $m$ machines and provides a set $A^{\mathrm{gd}}[m, \ell]$ with $\lambda$-approximation guarantee to the optimal solution $A^{\mathrm{oc}}[\ell]$, i.e., $f(A^{\mathrm{gd}}[m, \ell]) \geq \lambda f(A^{\mathrm{oc}}[\ell])$. Let us assume that we could run DISCARD with the unknown value $\ell = k$. Then the solution we get satisfies $f(A^{\mathrm{gd}}[m, k]) \geq \lambda Q$. Thus, we are not guaranteed to get $Q$ anymore. Now, what we can do (still under the assumption that we know $k$) is to repeatedly run DISCARD in order to augment our solution set until we get the desired value $Q$. Note that for each invocation of DISCARD, to find a set of size $\ell = k$, we have to take into account the solutions $A$ that we have accumulated so far. So,

| **Algorithm 1** Approximate Submodular Cover | **Algorithm 2** Approximate OPTCARD |
|---|---|
| **Input:** Set $V$, constraint $Q$.<br>**Output:** Set $A$.<br><br>1: $\ell = 1$.<br><br>2: $A^{\text{oc}}[\ell] = \text{OPTCARD}(V, \ell)$.<br><br>3: **while** $f(A^{\text{oc}}[\ell]) < Q$ **do**<br><br>4: $\quad \ell = \ell \times 2$.<br><br>5: $\quad A^{\text{oc}}[l] = \text{OPTCARD}(V, \ell)$.<br><br>6: $A = A^{\text{oc}}[\ell]$.<br><br>7: Return $A$. | **Input:** Set $V$, #of partitions $m$, constraint $Q$, $\ell$.<br>**Output:** Set $A^{\text{dc}}[m]$.<br>1: $r = 0$, $A^{\text{gd}}[m, \ell] = \emptyset$, .<br>2: **while** $f(A^{\text{gd}}[m, \ell]) < Q$ **do**<br>3: $\quad A = A^{\text{gd}}[m, \ell]$.<br>4: $\quad r = r + 1$.<br>5: $\quad A^{\text{gd}}[m, \ell] = \text{DISCARD}(V, m, \ell, A)$.<br>6: $\quad$ **if** $f(A^{\text{gd}}[m, \ell]) - f(A) \geq \lambda(Q - f(A))$ **then**<br>7: $\quad\quad A^{\text{dc}}[m] = \{A^{\text{gd}}[m, \ell] \cup A\}$.<br>8: $\quad$ **else**<br>9: $\quad\quad$ break<br>10: Return $A^{\text{dc}}[m]$. |

by overloading the notation, $\text{DISCARD}(V, m, \ell, A)$ returns a set of size $\ell$ given that $A$ has already been selected in previous rounds (i.e., DISCARD computes the marginal gains w.r.t. $A$). Note that at every invocation –thanks to submodularity– DISCARD increases the value of the solution by at least $\lambda(Q - f(A))$. Therefore, by running DISCARD at most $\lceil \log(Q)/\lambda \rceil$ times we get $Q$.

Unfortunately, we do not know the optimum value $k$. So, we can feed an estimate $\ell$ of the size of the optimum solution $k$ to DISCARD. Now, again thanks to submodularity, DISCARD can check whether this $\ell$ is good enough or not: if the improvement in the value of the solution is not at least $\lambda(Q - f(A))$ during the augmentation process, we can infer that $\ell$ is a too small estimate of $k$ and we cannot get the desired value $Q$ by using $\ell$ – so we apply the doubling strategy again.

**Theorem 4.1.** *Let* DISCARD *be a distributed algorithm for cardinality-constrained submodular maximization with $\lambda$ approximation guarantee. Then, Algorithm 1 (where* OPTCARD *is replaced with Approximate* OPTCARD, *Algorithm 2) runs in at most $\lceil \log(k) + \log(Q)/\lambda + 1 \rceil$ rounds and produces a solution of size at most $\lceil 2k + 2\log(Q)k/\lambda \rceil$.*

### 4.3 Trading Off Communication Cost and Number of Rounds

While Algorithm 1 successfully finds a distributed solution $A^{\text{dc}}[m]$ with $f(A^{\text{dc}}[m]) \geq Q$, (c.f. 4.1), the intermediate problem instances (i.e., invocations of DISCARD) are required to select sets of size up to twice the size of the optimal solution $k$, and these solutions are communicated between all machines. Oftentimes, $k$ is quite large and we do not want to have such a large communication cost per round. Now, instead of finding an $\ell \geq k$ what we can do is to find a smaller $\ell \geq \alpha k$, for $0 < \alpha \leq 1$ and augment these smaller sets in each round of Algorithm 2. This way, the communication cost reduces to an $\alpha$ fraction (per round), while the improvement in the value of the solution is at least $\alpha\lambda(Q - f(A^{\text{gd}}[m, \ell]))$. Consequently, we can trade-off the communication cost per round with the total number of rounds. As a positive side effect, for $\alpha < 1$, since in each invocation of DISCARD it returns smaller sets, the final solution set size can potentially get closer to the optimum solution size $k$. For instance, for the extreme case of $\alpha = 1/k$ we recover the solution of the sequential greedy algorithm (up to $O(1/\lambda)$). We see this effect in our experimental results.

### 4.4 DISCOVER

The DISCOVER algorithm is shown in Algorithm 3. The algorithm proceeds in rounds, with communication between machines taking place only between successive rounds. In particular, DISCOVER takes the ground set $V$, the number of partitions $m$, and the trade-off parameter $\alpha$. It starts with $\ell = 1$, and $A^{\text{dc}}[m] = \emptyset$. It then augments the set $A^{\text{dc}}[m]$ with set $A^{\text{gd}}[m, \ell]$ of at most $\ell$ new elements using an arbitrary distributed algorithm for submodular maximization under cardinality constraint, DISCARD. If the gain from adding $A^{\text{gd}}[m, \ell]$ to $A^{\text{dc}}[m]$ is at least $\alpha\lambda(Q - f(A^{\text{gd}}[m, \ell]))$, then we continue augmenting $A^{\text{gd}}[m, \ell]$ with another set of at most $\ell$ elements. Otherwise, we double $\ell$ and restart the process with $2\ell$. We repeat this process until we get $Q$.

**Theorem 4.2.** *Let* DISCARD *be a distributed algorithm for cardinality-constrained submodular maximization with $\lambda$ approximation guarantee. Then,* DISCOVER *runs in at most $\lceil \log(\alpha k) + \log(Q)/(\lambda\alpha) + 1 \rceil$ rounds and produces a solution of size $\lceil 2\alpha k + \log(Q)2k/\lambda \rceil$.*

**Algorithm 3** DISCOVER

**Input:** Set $V$, #of partitions $m$, constraint $Q$, trade off parameter $\alpha$.
**Output:** Set $A^{\mathrm{dc}}[m]$.
1: $A^{\mathrm{dc}}[m] = \emptyset$, $r = 0$.
2: **while** $f(A^{\mathrm{dc}}[m]) < Q$ **do**
3:      $r = r + 1$.
4:      $A^{\mathrm{gd}}[m, \ell] = \mathrm{DISCARD}(V, m, \ell, A^{\mathrm{dc}}[m])$.
5:      **if** $f(A^{\mathrm{dc}}[m] \cup A^{\mathrm{gd}}[m, \ell]) - f(A^{\mathrm{dc}}[m]) \geq \alpha\lambda(Q - f(A^{\mathrm{dc}}[m]))$ **then**
6:         $A^{\mathrm{dc}}[m] = \{A^{\mathrm{dc}}[m] \cup A^{\mathrm{gd}}[m, \ell]\}$.
7:      **else**
8:         $\ell = \ell \times 2$.
9: Return $A^{\mathrm{dc}}[m]$.

**GREEDI as Subroutine:** So far, we have assumed that a distributed algorithm DISCARD that runs on $m$ machines is given to us as a black box, which can be used to find sets of cardinality $\ell$ and obtain a $\lambda$-factor of the optimal solution. More concretely, we can use GREEDI, a recently proposed distributed algorithm for maximizing submodular functions under a cardinality constraint [17] (outlined in Algorithm 4). It first distributes the ground set $V$ to $m$ machines. Then each machine $i$ separately runs the standard greedy algorithm to produce a set $A_i^{\mathrm{gc}}[\ell]$ of size $\ell$. Finally, the solutions are merged, and another round of greedy selection is performed (over the merged results) in order to return the solution $A^{\mathrm{gd}}[m, \ell]$ of size $\ell$. It was proven that GREEDI provides a $(1 - e^{-1})^2 / \min(m, \ell)$-approximation to the optimal solution [17]. Here, we prove a (tight) improved bound on the performance of GREEDI. More formally, we have the following theorem.

**Theorem 4.3.** *Let $f$ be a monotone submodular function and let $\ell > 0$. Then, GREEDI produces a solution $A^{\mathrm{gd}}[m, \ell]$ where $f(A^{\mathrm{gd}}[m, \ell]) \geq \frac{1}{36\sqrt{\min(m,\ell)}} f(A^{\mathrm{c}}[\ell])$.*

**Algorithm 4** Greedy Distributed Submodular Maximization (GREEDI)

**Input:** Set $V$, #of partitions $m$, constraint $\ell$.
**Output:** Set $A^{\mathrm{gd}}[m, \ell]$.
1: Partition $V$ into $m$ sets $V_1, V_2, \ldots, V_m$.
2: Run the standard greedy algorithm on each set $V_i$. Find a solution $A_i^{\mathrm{gc}}[\ell]$.
3: Merge the resulting sets: $B = \cup_{i=1}^m A_i^{\mathrm{gc}}[\ell]$.
4: Run the standard greedy algorithm on $B$ until $\ell$ elements are selected. Return $A^{\mathrm{gd}}[m, \ell]$.

We illustrate the resulting algorithm DISCOVER using GREEDI as subroutine in Figure 1. By combining Theorems 4.2 and 4.3, we will have the following.

**Corollary 4.4.** *By using GREEDI, we get that DISCOVER produces a solution of size $\lceil 2\alpha k + 72\log(Q)k\sqrt{\min(m, \alpha k)}\rceil$ and runs in at most $\lceil \log(\alpha k) + 36\sqrt{\min(m, \alpha k)}\log(Q)/\alpha + 1\rceil$ rounds.*

Note that for a constant number of machines $m$, $\alpha = 1$ and a large solution size $\alpha k \geq m$, the above result simply implies that in at most $O(\log(kQ))$ rounds, DISCOVER produces a solution of size $O(k\log Q)$. In contrast, the greedy solution with $O(k\log Q)$ rounds (which is much larger than $O(\log(kQ))$) produces a solution of the same quality.

Very recently, a $(1 - e^{-1})/2$-approximation guarantee was proven for the randomized version of GREEDI [26, 25]. This suggests that, if it is possible to reshuffle (i.e., randomly re-distribute $V$ among the $m$ machines) the ground set each time that we revoke GREEDI, we can benefit from these stronger approximation guarantees (which are independent of $m$ and $k$). Note that Theorem 4.2 does not directly apply here, since it requires a deterministic subroutine for constrained submodular maximization. We defer the analysis to a longer version of this paper.

As a final technical remark, for our theoretical results to hold we have assumed that the utility function $f$ is integral. In some applications (like active set selection) this assumption may not hold. In these cases, either we can appropriately discretize and rescale the function, or instead of achieving

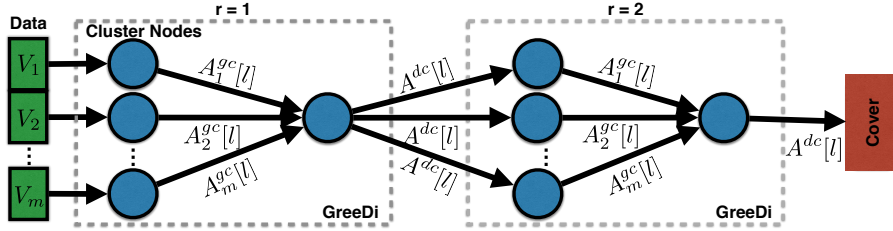

Figure 1: Illustration of our multi-round algorithm DISCOVER , assuming it terminates in two rounds (without doubling search for $\ell$).

the utility $Q$, try to reach $(1-\epsilon)Q$, for some $0 < \epsilon < 1$. In the latter case, we can simply replace $Q$ with $Q/\epsilon$ in Theorem 4.2.

## 5  Experiments

In our experiments we wish to address the following questions: 1) How well does DISCOVER perform compare to the centralized greedy solution; 2) How is the trade-off between the solution size and the number of rounds affected by parameter $\alpha$; and 3) How well does DISCOVER scale to massive data sets. To this end, we run DISCOVER on three scenarios: exemplar based clustering, active set selection in GPs, and vertex cover problem. For vertex cover, we report experiments on a large social graph with more than 65.6 million vertices and 1.8 billion edges. Since the constant in Theorem 4.3 is not optimized, we used $\lambda = 1/\sqrt{\min(m,k)}$ in all the experiments.

**Exemplar based Clustering.**   Our exemplar based clustering experiments involve DISCOVER applied to the clustering utility $f(S)$ described in Section 3.2 with $d(x,x') = \|x - x'\|^2$. We perform our experiments on a set of 10,000 *Tiny Images* [28]. Each 32 by 32 RGB pixel image is represented as a 3,072 dimentional vectors. We subtract from each vector the mean value, then normalize it to have unit norm. We use the origin as the auxiliary exemplar for this experiment. Fig. 2a compares the performance of our approach to the centralized benchmark with the number of machines set to $m = 10$ and varying coverage percentage $Q = (1-\epsilon)f(V)$. Here, we have $\beta = (1-\epsilon)$. It can be seen that DISCOVER provides a solution which is very close to the centralized solution, with a number of rounds much smaller than the solution size. Varying $\alpha$ results in a tradeoff between solution size and number of rounds.

**Active Set Selection.**   Our active set selection experiments involve DISCOVER applied to the log-determinant function $f(S)$ described in Section 3.2, using an exponential kernel $K(e_i, e_j) = \exp(-|e_i - e_j|^2/0.75)$. We use the *Parkinsons Telemonitoring* dataset [29] comprised of 5,875 biomedical voice measurements with 22 attributes from people in early-stage Parkinson's disease. Fig. 2b compares the performance of our approach to the benchmark with the number of machines set to $m = 6$ and varying coverage percentage $Q = (1-\epsilon)f(V)$. Again, DISCOVER performs close to the centralized greedy solution, even with very few rounds. Again we see a tradeoff by varying $\alpha$.

**Large Scale Vertex Cover with Spark.**   As our large scale experiment, we applied DISCOVER to the Friendster network consists of 65,608,366 nodes and 1,806,067,135 edges [30]. The average out-degree is 55.056 while the maximum out-degree is 5,214. The disk footprint of the graph is 30.7GB, stored in 246 part files on HDFS. Our experimental infrastructure was a cluster of 8 quad-core machines with 32GB of memory each, running Spark. We set the number of reducers to $m = 64$.

Each machine carried out a set of map/reduce tasks in sequence, where each map/reduce stage corresponds to running GREEDI with a specific values of $\ell$ on the whole data set. We first distributed the data uniformly at random to the machines, where each machine received $\approx$1,025,130 vertices ($\approx$12.5GB RAM). Then we start with $\ell = 1$, perform a map/reduce task to extract one element. We then communicate back the results to each machine and based on the improvement in the value of the solution, we perform another round of map/reduce calculation with either the the same value for $\ell$ or $2 \times \ell$. We continue performing map/reduce tasks until we get the desired value $Q$.

We examine the performance of DISCOVER by obtaining covers for 50%, 30%, 20% and 10% of the whole graph. The total running time of the algorithm for the above coverage percentages with $\alpha = 1$ was about 5.5, 1.5, 0.6 and 0.1 hours respectively. For comparison, we ran the centralized

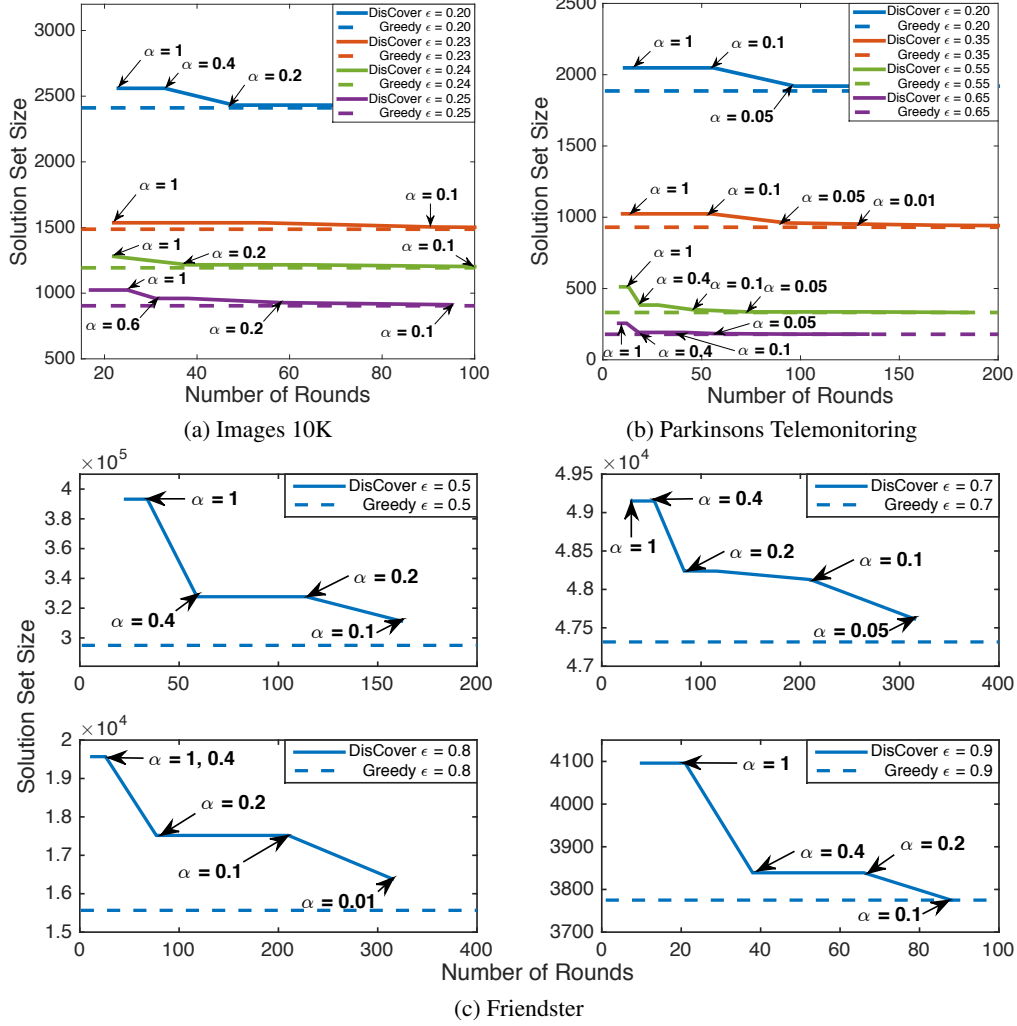

(a) Images 10K

(b) Parkinsons Telemonitoring

(c) Friendster

Figure 2: Performance of DISCOVER compared to the centralized solution. a, b) show the solution set size vs. the number of rounds for various $\alpha$, for a set of 10,000 *Tiny Images* and *Parkinsons Telemonitoring*. c) shows the same quantities for the Friendster network with 65,608,366 vertices.

greedy on a computer of 24 cores and 256GB memory. Note that, loading the entire data set into memory requires 200GB of RAM, and running the centralized greedy algorithm for 50% cover requires at least another 15GB of RAM. This highlights the challenges in applying the centralized greedy algorithm to larger scale data sets. Fig. 2c shows the solution set size versus the number of rounds for various $\alpha$ and different coverage constraints. We find that by decreasing $\alpha$, DISCOVER's solutions quickly converge (in size) to those obtained by the centralized solution.

# 6 Conclusion

We have developed the first efficient distributed algorithm –DISCOVER – for the submodular cover problem. We have theoretically analyzed its performance and showed that it can perform arbitrary close to the centralized (albeit impractical in context of large data sets) greedy solution. We also demonstrated the effectiveness of our approach through extensive experiments, including vertex cover on a graph with 65.6 million vertices using Spark. We believe our results provide an important step towards solving submodular optimization problems in very large scale, real applications.

**Acknowledgments.** This research was supported by ERC StG 307036, a Microsoft Faculty Fellowship and an ETH Fellowship.

## Footnotes

[1]Note that while reduction from submodular coverage to submodular maximization has been used (e.g., [27]), the straightforward application to the distributed setting incurs large communication cost.

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
