[Supplementary Material · distributed-cover-supp.pdf]

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

| **Output:** Set $A$. | **Output:** Set $A^{\mathrm{dc}}[m]$. |
| 1: $\ell = 1$. | 1: $r = 0$, $A^{\mathrm{gd}}[m,\ell] = \emptyset$, . |
| 2: $A^{\mathrm{oc}}[\ell] = \mathrm{OPTCARD}(V,\ell)$. | 2: **while** $f(A^{\mathrm{gd}}[m,\ell]) < Q$ **do** |
| 3: **while** $f(A^{\mathrm{oc}}[\ell]) < Q$ **do** | 3:    $A = A^{\mathrm{gd}}[m,\ell]$. |
| 4:    $\ell = \ell \times 2$. | 4:    $r = r + 1$. |
| 5:    $A^{\mathrm{oc}}[l] = \mathrm{OPTCARD}(V,\ell)$. | 5:    $A^{\mathrm{gd}}[m,\ell] = \mathrm{DISCARD}(V,m,\ell,A)$. |
| | 6:    **if** $f(A^{\mathrm{gd}}[m,\ell]) - f(A) \geq \lambda(Q - f(A))$ **then** |
| 6: $A = A^{\mathrm{oc}}[\ell]$. | 7:        $A^{\mathrm{dc}}[m] = \{A^{\mathrm{gd}}[m,\ell] \cup A\}$. |
| 7: Return $A$. | 8:    **else** |
| | 9:        break |
| | 10: Return $A^{\mathrm{dc}}[m]$. |

by overloading the notation, $\mathrm{DISCARD}(V,m,\ell,A)$ returns a set of size $\ell$ given that $A$ has already been selected in previous rounds (i.e., $\mathrm{DISCARD}$ computes the marginal gains w.r.t. $A$). Note that at every invocation –thanks to submodularity– $\mathrm{DISCARD}$ increases the value of the solution by at least $\lambda(Q - f(A))$. Therefore, by running $\mathrm{DISCARD}$ at most $\lceil \log(Q)/\lambda \rceil$ times we get $Q$.

Unfortunately, we do not know the optimum value $k$. So, we can feed an estimate $\ell$ of the size of the optimum solution $k$ to $\mathrm{DISCARD}$. Now, again thanks to submodularity, $\mathrm{DISCARD}$ can check whether this $\ell$ is good enough or not: if the improvement in the value of the solution is not at least $\lambda(Q - f(A))$ during the augmentation process, we can infer that $\ell$ is a too small estimate of $k$ and we cannot get the desired value $Q$ by using $\ell$ – so we apply the doubling strategy again.

**Theorem 4.1.** *Let* $\mathrm{DISCARD}$ *be a distributed algorithm for cardinality-constrained submodular maximization with* $\lambda$ *approximation guarantee. Then, Algorithm 1 (where* $\mathrm{OPTCARD}$ *is replaced with Approximate* $\mathrm{OPTCARD}$, *Algorithm 2) runs in at most* $\lceil \log(k) + \log(Q)/\lambda + 1 \rceil$ *rounds and produces a solution of size at most* $\lceil 2k + 2\log(Q)k/\lambda \rceil$.

### 4.3 Trading Off Communication Cost and Number of Rounds

While Algorithm 1 successfully finds a distributed solution $A^{\mathrm{dc}}[m]$ with $f(A^{\mathrm{dc}}[m]) \geq Q$, (c.f. 4.1), the intermediate problem instances (i.e., invocations of $\mathrm{DISCARD}$) are required to select sets of size up to twice the size of the optimal solution $k$, and these solutions are communicated between all machines. Oftentimes, $k$ is quite large and we do not want to have such a large communication cost per round. Now, instead of finding an $\ell \geq k$ what we can do is to find a smaller $\ell \geq \alpha k$, for $0 < \alpha \leq 1$ and augment these smaller sets in each round of Algorithm 2. This way, the communication cost reduces to an $\alpha$ fraction (per round), while the improvement in the value of the solution is at least $\alpha\lambda(Q - f(A^{\mathrm{gd}}[m,\ell]))$. Consequently, we can trade-off the communication cost per round with the total number of rounds. As a positive side effect, for $\alpha < 1$, since in each invocation of $\mathrm{DISCARD}$ it returns smaller sets, the final solution set size can potentially get closer to the optimum solution size $k$. For instance, for the extreme case of $\alpha = 1/k$ we recover the solution of the sequential greedy algorithm (up to $O(1/\lambda)$). We see this effect in our experimental results.

### 4.4 DISCOVER

The $\mathrm{DISCOVER}$ algorithm is shown in Algorithm 3. The algorithm proceeds in rounds, with communication between machines taking place only between successive rounds. In particular, $\mathrm{DISCOVER}$ takes the ground set $V$, the number of partitions $m$, and the trade-off parameter $\alpha$. It starts with $\ell = 1$, and $A^{\mathrm{dc}}[m] = \emptyset$. It then augments the set $A^{\mathrm{dc}}[m]$ with set $A^{\mathrm{gd}}[m,\ell]$ of at most $\ell$ new elements using an arbitrary distributed algorithm for submodular maximization under cardinality constraint, $\mathrm{DISCARD}$. If the gain from adding $A^{\mathrm{gd}}[m,\ell]$ to $A^{\mathrm{dc}}[m]$ is at least $\alpha\lambda(Q - f(A^{\mathrm{gd}}[m,\ell]))$, then we continue augmenting $A^{\mathrm{gd}}[m,\ell]$ with another set of at most $\ell$ elements. Otherwise, we double $\ell$ and restart the process with $2\ell$. We repeat this process until we get $Q$.

**Theorem 4.2.** *Let* $\mathrm{DISCARD}$ *be a distributed algorithm for cardinality-constrained submodular maximization with* $\lambda$ *approximation guarantee. Then,* $\mathrm{DISCOVER}$ *runs in at most* $\lceil \log(\alpha k) + \log(Q)/(\lambda\alpha) + 1 \rceil$ *rounds and produces a solution of size* $\lceil 2\alpha k + \log(Q)2k/\lambda \rceil$.

**Algorithm 3** DISCOVER

**Input:** Set $V$, #of partitions $m$, constraint $Q$, trade off parameter $\alpha$.
**Output:** Set $A^{\text{dc}}[m]$.
 1: $A^{\text{dc}}[m] = \emptyset$, $r = 0$.
 2: **while** $f(A^{\text{dc}}[m]) < Q$ **do**
 3:     $r = r + 1$.
 4:     $A^{\text{gd}}[m, \ell] = \text{DISCARD}(V, m, \ell, A^{\text{dc}}[m])$.

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

# Appendix

This section presents the complete proofs of theorems presented in the article.

### Proof of Theorem 4.1

This theorem is a special case of theorem 4.2 with $\alpha = 1$.

### Proof of Theorem 4.2

We first prove a lemma which will be used in the proof of the theorem.

**Lemma 6.1.** *Let $\ell^*$ be the final value of $\ell$ at the end of the DISCOVER Algorithm. Then,*

$$\ell^* < 2\alpha k.$$

**Proof:** We consider it in two cases. In first case $\ell^* < \alpha k$ in which case the lemma is trivially true. Else at some iteration of DISCOVER we have that $\alpha k \leq \ell < 2\alpha k$. We then show that for such a value of $\ell$ step 5 always holds and hence $\ell$ is never incremented, proving the lemma. For the rest of the proof consider such a value of $\ell$.

Let $A^{\text{gd}}[m, \ell]$ be the set returned by GREEDI when requested to return a solution of size $\ell$ when the current solution is $A^{\text{dc}}[m]$. Let $S_p^*$ be a set of size $p$ which maximizes $f(S_p^* \cup A^{\text{dc}}[m])$. Let $S_k^* = \{e_1, e_2, \ldots, e_k\}$ be the elements when then are sorted according to their marginal contributions when they are added one by one to $A^{\text{dc}}[m]$. Let $S_k^*(l) = \{e_1, e_2, \ldots, e_l\}$. By definition of sorting according to decreasing marginal values we have the following inequality

$$\forall p, f(S_k^*(p+1) \cup A^{\text{dc}}[m]) - f(S_k^*(p) \cup A^{\text{dc}}[m]) \geq f(S_k^*(p+2) \cup A^{\text{dc}}[m]) - f(S_k^*(p+1) \cup A^{\text{dc}}[m]) \tag{3}$$

Now we have the following set of inequalities.

$$
\begin{aligned}
& f(A^{\text{gd}}[m, \ell] \cup A^{\text{dc}}[m]) - f(A^{\text{dc}}[m]) && \\
& \geq \lambda(f(S_\ell^* \cup A^{\text{dc}}[m]) - f(A^{\text{dc}}[m])) && \text{By definition of } \lambda \\
& \geq \lambda(f(S_{\alpha k}^* \cup A^{\text{dc}}[m]) - f(A^{\text{dc}}[m])) && \text{By monotonicity} \\
& \geq \lambda(f(S_k^*(\alpha k) \cup A^{\text{dc}}[m]) - f(A^{\text{dc}}[m])) && \text{By definition of } S_{\alpha k}^* \\
& = \lambda \sum_{i=0}^{\alpha k - 1}(f(S_k^*(i+1) \cup A^{\text{dc}}[m]) - f(S_k^*(i) \cup A^{\text{dc}}[m])) && \\
& \geq \lambda \alpha \sum_{i=0}^{k-1}(f(S_k^*(i+1) \cup A^{\text{dc}}[m]) - f(S_k^*(i) \cup A^{\text{dc}}[m])) && \text{Elements sorted by marginal values} \\
& = \lambda \alpha(f(S_k^* \cup A^{\text{dc}}[m]) - f(A^{\text{dc}}[m])) && \\
& \geq \lambda \alpha(f(S_k^*) - f(A^{\text{dc}}[m])) && \text{By monotonicity} \\
& \geq \lambda \alpha(Q - f(A^{\text{dc}}[m])) && \text{By monotonicity}
\end{aligned}
$$

The above set of inequalities prove that step 5 always holds and hence $\ell$ is never incremented once it reaches a value such that $\alpha k \leq \ell < 2\alpha k$, proving the lemma. $\qquad\square$

Now armed with Lemma 6.1 we complete the proof of theorem 4.2. Consider step 8 of DISCOVER. This step can get executed at most $\log(2\alpha k)$ times because of lemma 6.1. Let $T_i$ be the solution set $S$ returned when step 5 of DISCOVER is satisfied for the $i^{th}$ time. Let $T^* = T_z$ be the solution before condition on step 5 is satisfied for the last time and condition on step 5 be satisfied $z + 1$

times. Then by the inequality in step 5 we have the inequality below.

$$
\begin{aligned}
f(T_{i+1}) - f(T_i) &\geq & \alpha\lambda(Q - f(T_i)) \\
\Rightarrow Q - f(T_{i+1}) &\leq & (1 - \alpha\lambda)(Q - f(T_i)) \\
\Rightarrow Q - f(T^*) &\leq & (1 - \alpha\lambda)^z Q \quad < 1 \\
\Rightarrow (1 - \alpha\lambda)^z &< & 1/Q \\
\Rightarrow z \log(1 - \alpha\lambda) &< & -\log(Q) \\
\Rightarrow z\alpha\lambda &> & \log(Q) \qquad \text{Since } \log(1 - x) \leq -x \\
\Rightarrow z &> & \tfrac{1}{\alpha\lambda} \log(Q)
\end{aligned}
$$

Hence the total number of rounds of the algorithm is $\log(\alpha k) + \frac{1}{\alpha\lambda}\log(Q) + 1$. Additionally since we pick at most $2\alpha k$ elements in each round satisfying step 5 of DISCOVER we get the solution is of size at most $2\alpha k \cdot (\frac{1}{\alpha\lambda}\log(Q) + 1) = 2\alpha k + \frac{2k}{\lambda}\log(Q)$.

**Proof of Theorem 4.3**

Let $\mathcal{G}(T, k)$ be the set returned by running greedy to choose k elements on $T$.

**Fact 6.2.** *Let $\Omega$ be any ground set and $S = \mathcal{G}(\Omega, k)$ and $O$ be any set such that $|O| \leq k$. Then we have the following inequality*

$$
f(S) \geq \frac{k}{|O|} \left( f(S \cup O) - f(S) \right) \tag{4}
$$

**Proof:** Let $S = \{e_1, e_2, \ldots, e_k\}$ sorted by the order in which greedy algorithm picks the elements. Let for each $i$, $S_i = \{e_1, e_2, \ldots, e_i\}$. Let $O = \{o_1, o_2, \ldots o_{|O|}\}$ and $O_i = \{o_1, o_2, \ldots, o_i\}$. Follows easily from the following set of equations

$$
\begin{aligned}
f(S) &= & \sum_{i=1}^{k} f(S_i) - f(S_{i-1}) & \\
&\geq & \sum_{i=1}^{k} \frac{1}{|O|} \sum_{j=1}^{|O|} f(S_{i=1} \cup \{o_j\}) - f(S_{i-1}) & \text{By property of greedy algorithm} \\
&\geq & \sum_{i=1}^{k} \frac{1}{|O|} \sum_{j=1}^{|O|} f(S_{i=1} \cup O_j) - f(S_{i-1} \cup O_{j-1}) & \text{By submodularity} \\
&= & \sum_{i=1}^{k} \frac{1}{|O|} (f(S_{i=1} \cup O) - f(S_{i-1})) & \\
&\geq & \sum_{i=1}^{k} \frac{1}{|O|} (f(S \cup O) - f(S)) & \text{By submodularity} \\
&= & \frac{k}{|O|} (f(S \cup O) - f(S)) &
\end{aligned}
$$

$\square$

Now we complete proof of theorem 4.3. Let $V_i$ be the set of elements on machine $i$. Let $S_i = \mathcal{G}(V_i, k)$. Then we will show a set $S \subseteq \cup_{i=1}^{m} S_i$ such that $|S| \leq k$ and $f(S) \geq \frac{1}{18\sqrt{\min(k,m)}} f(\mathrm{OPT})$.

Then the theorem follows because $\mathcal{G}(\cup_{i=1}^{m} S_i, k)$ produces at least a $1 - 1/e$ approximation to $S$.

Let $S_1, S_2, \ldots, S_q$ be the sets such that $S_i \cap \mathrm{OPT} \neq \emptyset$. Then note that $q \leq \min(k, m)$. We will restrict the analysis to these sets. Let $\mathrm{OPT}_i = V_i \cap \mathrm{OPT}$. We construct the set $S$ by iteratively processing each $S_i$ and adding elements $S$. Let $L_i$ be the set $S$ after we process $S_i$. We start with $L_0 = \emptyset$. Let us say we have processed till $S_{j-1}$, we will show what happens when we process $S_j$.

Let $S_j = \{s_j^1, s_j^2, \ldots, s_j^k\}$ in the order they are taken by the greedy algorithm. Let $S_j^p$ be the set of first $p$ elements in $S_j$. Then there are three cases.

1. Assume for each $p \leq k$ the following inequality is satisfied.

$$
f(L_{j-1} \cup S_j^p \cup \mathrm{OPT}_j) - f(L_{j-1} \cup S_j^p) \geq \frac{1}{2} \left( f(L_{j-1} \cup \mathrm{OPT}_j) - f(L_{j-1}) \right) \tag{5}
$$

Then choose $|\text{OPT}_j|$ elements from $S_j$ greedily and add them to $L_{j-1}$. Then we prove a lower bound on the improvement.

$$f(L_j) - f(L_{j-1}) \tag{6}$$

$$\geq \frac{|\text{OPT}_j|}{k}\left(f(L_j \cup S_j) - f(L_j)\right) \qquad\qquad \text{Because of greedy algorithm}$$

$$\geq \frac{|\text{OPT}_j|}{k}f(S_j) - \frac{|\text{OPT}_j|}{k}f(L_j) \qquad\qquad \text{Because of monotonicity}$$

$$\geq f(S_j \cup \text{OPT}_j) - f(S_j) - \frac{|\text{OPT}_j|}{k}f(L_j) \qquad\qquad \text{Because of fact 6.2}$$

$$\geq f(L_{j-1} \cup S_j \cup \text{OPT}_j) - f(L_{j-1} \cup S_j) - \frac{|\text{OPT}_j|}{k}f(L_j) \qquad\qquad \text{Because of submodularity}$$

$$\geq \frac{1}{2}\left(f(L_{j-1} \cup \text{OPT}_j) - f(L_{j-1})\right) - \frac{|\text{OPT}_j|}{k}f(L_j) \qquad\qquad \text{Because of assumption in equation 5}$$

$$\geq \frac{1}{2}\left(f(L_l \cup \text{OPT}_j) - f(L_l)\right) - \frac{|\text{OPT}_j|}{k}f(L_j) \qquad\qquad \text{by submodularity}$$

$$\geq \frac{1}{2}\left(f(L_l \cup \text{OPT}_j) - f(L_l)\right) - \frac{|\text{OPT}_j|}{k}f(L_l) \qquad\qquad \text{by monotonicity}$$

$$\tag{7}$$

2. The next two cases are folded into this case. Let $c(j)$ be the minimum index such that the following inequality is satisfied.

$$f(L_{j-1} \cup S_j^{c(j)} \cup \text{OPT}_j) - f(L_{j-1} \cup S_j^{c(j)}) \leq \frac{1}{2}\left(f(L_{j-1} \cup \text{OPT}_j) - f(L_{j-1})\right) \quad (8)$$

Then we get the following sequence of inequalities

•
$$f(L_{j-1} \cup S_j^{c(j)}) - f(L_{j-1}) \tag{9}$$

$$= f(L_{j-1} \cup S_j^{c(j)} \cup \text{OPT}_j) - f(L_{j-1}) - \left(f(L_{j-1} \cup S_j^{c(j)} \cup \text{OPT}_j) - f(L_{j-1} \cup S_j^{c(j)})\right)$$

$$\geq f(L_{j-1} \cup \text{OPT}_j) - f(L_{j-1}) - \left(f(L_{j-1} \cup S_j^{c(j)} \cup \text{OPT}_j) - f(L_{j-1} \cup S_j^{c(j)})\right)$$

$$\text{Because of monotonicity}$$

$$\geq f(L_{j-1} \cup \text{OPT}_j) - f(L_{j-1}) - \frac{1}{2}\left(f(L_{j-1} \cup \text{OPT}_j) - f(L_{j-1})\right)$$

$$\text{From equation 8}$$

$$= \frac{1}{2}\left(f(L_{j-1} \cup \text{OPT}_j) - f(L_{j-1})\right) \tag{10}$$

• Consider when $c(j) > 1$. Then note that for any $p < c(j)$ we have that $f(L_{j-1} \cup S_j^p \cup \text{OPT}_j) - f(L_{-1}j \cup S_j^p) \geq \frac{1}{2}\left(f(L_{j-1} \cup \text{OPT}_j) - f(L_{j-1})\right)$. Hence we also get the following inequality

$$f(S_j^{p+1}) - f(S_j^p) \geq \frac{1}{|\text{OPT}_j|} \sum_{e \in \text{OPT}_j} f(S_j^p \cup \{e\}) - f(S_j^p) \qquad\qquad \text{By choice of greedy}$$

$$\geq \frac{1}{|\text{OPT}_j|}\left(f(S_j^p \cup \text{OPT}_j) - f(S_j^p)\right) \qquad\qquad \text{By submodularity}$$

$$\geq \frac{1}{|\text{OPT}_j|}\left(f(L_{j-1} \cup S_j^p \cup \text{OPT}_j) - f(L_{j-1} \cup S_j^p)\right) \qquad\qquad \text{By submodularity}$$

$$\geq \left(\frac{1}{2|\text{OPT}_j|}\right)\left(f(L_{j-1} \cup \text{OPT}_j) - f(L_{j-1})\right) \qquad\qquad \text{By assumption}$$

$$\Rightarrow f(S_j^{c(j)}) \geq \left(\frac{c(j)}{2|\text{OPT}_j|}\right)\left(f(L_{j-1} \cup \text{OPT}_j) - f(L_{j-1})\right) \qquad\qquad \text{Summing above inequalities}$$

$$\tag{11}$$

$$\tag{12}$$

Then we have three different situations

(a) $c(j) <= \sqrt{q} \cdot |\text{OPT}_j|$ and $c(j) > \text{OPT}_j$ then add $|\text{OPT}_j|$ elements to $L_{j-1}$ greedily from $S_j$. Then we get

$$f(L_j) - f(L_{j-1}) \tag{13}$$

$$\geq \frac{|\text{OPT}_j|}{c(j)} \left( f(L_j \cup S_j^{c(j)}) - f(L_j) \right) \qquad \text{from property of greedy algorithm}$$

$$\geq \frac{|\text{OPT}_j|}{2c(j)} \left( f(L_{j-1} \cup \text{OPT}_j) - f(L_{j-1}) \right) \qquad \text{from equation 10}$$

$$\geq \frac{1}{2\sqrt{q}} \left( f(L_{j-1} \cup \text{OPT}_j) - f(L_{j-1}) \right) \qquad \text{because of assumption } c(j) <= \sqrt{q} \cdot |\text{OPT}_j|$$

$$\geq \frac{1}{2\sqrt{q}} \left( f(L_q \cup \text{OPT}_j) - f(L_q) \right) \qquad \text{from submodularity}$$

$$\tag{14}$$

(b) $c(j) > \sqrt{q} \cdot |\text{OPT}_j|$ then add $|\text{OPT}_j|$ elements to $L_{j-1}$ greedily from $S_j$. Then we get

$$f(L_j) - f(L_{j-1}) \tag{15}$$

$$\geq \frac{|\text{OPT}_j|}{c(j)} \left( f(L_j \cup S_j^{c(j)}) - f(L_j) \right) \qquad \text{from property of greedy algorithm}$$

$$\geq \frac{|\text{OPT}_j|}{c(j)} \left( f(S_j^{c(j)}) - f(L_j) \right) \qquad \text{from monotonicity}$$

$$\geq \frac{1}{2} \left( f(L_{j-1} \cup \text{OPT}_j) - f(L_{j-1}) \right) - \frac{|\text{OPT}_j|}{c(j)} f(L_j) \qquad \text{from equation 12}$$

$$\geq \frac{1}{2} \left( f(L_{j-1} \cup \text{OPT}_j) - f(L_{j-1}) \right) - \frac{1}{\sqrt{q}} f(L_j) \qquad \text{from assumption } c(j) > \sqrt{q} \cdot |\text{OPT}_j|$$

$$\geq \frac{1}{2} \left( f(L_q \cup \text{OPT}_j) - f(L_q) \right) - \frac{1}{\sqrt{q}} f(L_j) \qquad \text{from submodularity}$$

$$\geq \frac{1}{2} \left( f(L_l \cup \text{OPT}_j) - f(L_l) \right) - \frac{1}{\sqrt{q}} f(L_l) \qquad \text{from monotonicity}$$

$$\tag{16}$$

(c) If $c(j) <= |\text{OPT}_j|$ then $L_j = L_{j-1} \cup S_j^{c(j)}$ and we get $f(L_j) - f(L_{j-1}) \geq \frac{1}{2} \left( f(L_{j-1} \cup \text{OPT}_j) - f(L_{j-1}) \right)$ from equation 10

We complete the proof by considering three different cases and proving the theorem in each of the three cases. For simplicity let the indices which satisfy condition one be $I_1$, condition 2a) (or condition 2c) be $I_{2a}$ and condition 2b) be $I_{2b}$. Let $\text{OPT}' = \cup_{i \in I_1} \text{OPT}_i$, $\text{OPT}'' = \cup_{i \in I_{2a}} \text{OPT}_i$ and $\text{OPT}''' = \cup_{i \in I_{2b}} \text{OPT}_i$. Then by simply submodularity we know that $\max(f(\text{OPT}'), f(\text{OPT}''), f(\text{OPT}''')) \geq f(\text{OPT})/3$. We deal with each case separately.

- Case 1 when $f(\textsc{opt}') \geq f(\textsc{opt})/3$

$$
\begin{aligned}
f(L_l) - f(\emptyset) &\geq \sum_{i \in I_1} f(L_i) - f(L_{i-1}) \\
&\geq \sum_{i \in I_1} \left( \frac{1}{2} \left( f(L_l \cup \textsc{opt}_j) - f(L_l) \right) - \frac{|\textsc{opt}_j|}{k} f(L_l) \right) \\
\Rightarrow 2 f(L_l) &\geq \sum_{i \in I_1} \frac{1}{2} \left( f(L_l \cup \textsc{opt}_j) - f(L_l) \right) && \text{Rearranging terms} \\
&\geq \frac{1}{2} \left( f(L_l \cup \textsc{opt}') - f(L_l) \right) && \text{By submodularity} \\
&\geq \frac{1}{2} \left( f(\textsc{opt}') - f(L_l) \right) && \text{By monotonicity} \\
\Rightarrow f(L_l) &\geq \frac{1}{5} f(\textsc{opt}') && \text{Rearranging terms} \\
&\geq \frac{1}{15} f(\textsc{opt}) && \text{By assumption}
\end{aligned}
\tag{17}
$$

- Case 2 when $f(\textsc{opt}'') \geq f(\textsc{opt})/3$

$$
\begin{aligned}
f(L_l) - f(\emptyset) &\geq \sum_{i \in I_{2a}} f(L_i) - f(L_{i-1}) \\
&\geq \sum_{i \in I_{2a}} \frac{1}{2\sqrt{q}} \left( f(L_l \cup \textsc{opt}_j) - f(L_l) \right) \\
&\geq \frac{1}{2\sqrt{q}} \left( f(L_l \cup \textsc{opt}'') - f(L_l) \right) && \text{By submodularity} \\
&\geq \frac{1}{2\sqrt{q}} O \left( f(\textsc{opt}'') - f(L_l) \right) && \text{By monotonicity} \\
\Rightarrow f(L_l) &\geq \frac{1}{4\sqrt{q}} f(\textsc{opt}'') && \text{Rearranging terms} \\
&\geq \frac{1}{12\sqrt{q}} f(\textsc{opt}) && \text{By assumption} \tag{18}
\end{aligned}
$$

- Case 3 when $f(\textsc{opt}''') \geq f(\textsc{opt})/3$

$$
\begin{aligned}
f(L_l) - f(\emptyset) &\geq \sum_{i \in I_{2b}} f(L_i) - f(L_{i-1}) \\
&\geq \sum_{i \in I_{2b}} \left( \frac{1}{2} \left( f(L_l \cup \textsc{opt}_j) - f(L_l) \right) - \frac{1}{\sqrt{q}} f(L_l) \right) \\
\Rightarrow (\sqrt{q} + 1) f(L_l) &\geq \sum_{I_{2b}} \frac{1}{2} \left( f(L_l \cup \textsc{opt}_j) - f(L_l) \right) && \text{Rearranging terms} \\
&\geq \frac{1}{2} \left( f(L_l \cup \textsc{opt}''') - f(L_l) \right) && \text{By submodularity} \\
&\geq \frac{1}{2} \left( f(\textsc{opt}''') - f(L_l) \right) && \text{By monotonicity} \\
\Rightarrow f(L_l) &\geq \frac{1}{2(\sqrt{q} + 2)} f(\textsc{opt}''') && \text{Rearranging terms} \\
&\geq \frac{1}{6(\sqrt{q} + 2)} f(\textsc{opt}) && \text{By assumption} \\
&\geq \frac{1}{18\sqrt{q}} f(\textsc{opt}) && \tag{19}
\end{aligned}
$$

Remember that $q = \min(k, m)$ which completes the proof.