[Reviews · NeurIPS 2015]

Submitted by Assigned_Reviewer_1

The paper studies submodular cover in the MapReduce setting. In this problem, we want to find the smallest set of item S such that f(S) is at least a given value Q for submodular function f. This is closely related to the problem of maximizing f(S) subject to the size of S is at most k. In the sequential setting, both problems can be solved with the greedy algorithm. This paper gives a MapReduce-style algorithm for submodular cover via calls to the recent work on distributed greedy algorithms for constrained submodular maximization in a black box fashion.The algorithm works in a similar fashion to the greedy algorithm. The greedy algorithm iteratively picks one element at a time so if the optimal solution has size k, it might need k*log(Q) iterations. The key insight of the paper is that one can batch k iterations into 1 call to the existing distributed algorithm for submodular maximization with cardinality constraint. Thus, the algorithm is simply calling the existing algorithm log(Q) times. One can also trade off between the number of rounds and solution quality by picking fewer than k elements in each call.

The paper is well-written and easy to understand. The experiments show the algorithm works well compared with the centralized solution.

Specific comments:

Theorem 4.3 on tighter analysis of GREEDI seems similar to appendix A of [25]. It might be good to include some comment on this.

Initialization l=1 missing in Algorithm 3.
Summary: The paper gives the first MapReduce-style algorithm for submodular cover based on recent works on the closely related problem of constrained submodular maximization. The algorithm works well compared with the centralized solution in experiments.

Submitted by Assigned_Reviewer_2

This paper proposes a distributed algorithm DISCOVER for solving submodular set cover problem. The idea of DISCOVER is to iteratively run the GreeDI as an oracle to incrementally obtain a solution until its valuation is greater than the solution quality requirement. The algorithm can tradeoff between the number of iterations and the communication cost needed for each iteration. This method is also validated on large-scale data applications.

I think this paper studies an important problem and the proposed algorithm -- DISCOVER is novel under the distributed setting. The paper is well presented. The algorithm is complemented with theoretical justification and the empirical validation is also extensive.

But still, I hope the authors could address the some of my questions below.

Some questions and comments:

(1) The section about the tradeoff between communication cost and number of rounds is a bit confusing to me. It seems to me that by choosing smaller \alpha, it requires more rounds, but each round needs smaller amount of communication. I think the goal should be to find an \alpha such that the overall communication cost is minimized, instead of only looking at the cost per round. I think it would be interesting to have more discussion on how to choose \alpha in practice.

(2) Instead of setting \alpha as a constant between 0 and 1, it may be more efficient to set alpha adaptively as the algorithm proceeds. One possibility could be to set \alpha close to 1 at the first few rounds, and then gradually decreases \alpha to 0 towards the end.

(3) In Alg. 3, suppose we set \alpha = 1. I think the tradeoff between the communication cost and the number of rounds may also be achieved by replacing the doubling strategy (Line 8 of Alg. 3) with \ell = \ell * \beta, where \beta > 1 is a hyperparameter that controls the tradeoff. When \beta is only slightly larger than 1, the algorithm takes more rounds with the communication cost of each round being small.

(4) In the experiment section, what \lambda is used for each experiment?

Minor comment:

The bound in Theorem 4.3 may be further improved: the following paper shows a guarantee of

(1-1/e)/ (2\sqrt{k}) with k being the cardinality constraint for GreeDI and also shows that GreeDI is tight at 1/\sqrt(k).

The Power of Randomization Distributed Submodular Maximization on Massive Datasets http://arxiv.org/pdf/1502.02606v2.pdf

Summary: This paper proposes a distributed algorithm for solving submodular set cover problem. The algorithm can tradeoff between the number of iterations and the communication cost needed for each iteration. This method is also validated on large-scale data applications.

Submitted by Assigned_Reviewer_3

This paper addresses distributed submodular cover. It is built heavily upon the existing work [17] on cardinality constrained submodular cover, and as such it appears to be a bit incremental. Fortunately, the authors justify their design with theoretical analysis. The result is a distributed submodular cover method that can deal with large dataset. The experiments on several applications show promising results. Overall, this is a solid work. One concern is the additional value of the general submodular cover over the cardinality constrained submodular cover. A discussion on this issue may be interesting, especially with respect to its impact to practical applications.

Summary: This paper presented an approach to the submodular cover problem, which is able to distribute across a cluster of machines. The method is well justified and the experimental results on various applications confirmed the effectiveness of the proposed method.

Submitted by Assigned_Reviewer_4

The paper proposes a repeated application of a distributed constrained integral monotone sub modular function maximisation algorithm to solve the sub modular cover problem using MapReduce. The doubling trick is used to adjust the constraint until a high-enough utility is achieved. The distributed part of the algorithm is fully contained in the black-box distributed constrained max sub-routine.

Remarks ---------- The paper gives the example of a log-det probability used with DPP and so on, and claims that this is monotone, but I fail to see why. Also, I don't think the reference given on this talks about log-det at all.

How is "number of rounds" actually defined? How is it defined for the non-distributed variant? This can be inferred by the reader, but it'd be clearer to say it.

Minor remarks ----------------- MapReduce offer*s*

"In particular, there is no efficient way to identify the optimum subset Aoc[l] in set V , unless P=NP"

- I find this type of vague statement should be avoided: it is more precise to say that the problem is NP hard.

the submission did not include line numbers
Summary: This is an interesting problem, and multiple experiments are provided on medium-sized data.

A log factor approximation guarantee is provided, based on a reduction to sub-routines with proven guarantees.

Author Feedback
Author rebuttal: We kindly thank the reviewers for their careful review.

Reviewer 1
-Theorem 4.3 on tighter analysis of GREEDI seems similar to appendix A of [25].
The proof provided in our paper has been derived in parallel to that of [25]. Also our proof technique is very different and our bound is better if the number m of machines is bounded. We'll certainly add a reference to [25].

-Initialization l=1 missing in Algorithm 3.
Thanks for pointing this out.

Reviewer 2
-It seems to me that by choosing smaller \alpha, it requires more rounds, but each round needs smaller amount of communication. I think the goal should be to find an \alpha such that the overall communication cost is minimized, instead of only looking at the cost per round. I think it would be interesting to have more discussion on how to choose \alpha in practice.
The intuition is correct. We in fact considered optimizing \alpha by trading off number_of_rounds and solution_set_size, suitably weighted. However, in large datasets, k could be huge, and the two quantities are of very different orders of magnitude, as shown, e.g., in our experiments (Fig 2.c). Thus, the optimization problem may not always be meaningful. Moreover, considering the amount of communication, it's not always possible to use the optimum value of \alpha. In practice, \alpha depends very much on the available infrastructure.

-Instead of setting \alpha as a constant between 0 and 1, it may be more efficient to set alpha adaptively as the algorithm proceeds. One possibility could be to set \alpha close to 1 at the first few rounds, and then gradually decreases \alpha to 0 towards the end.
This is an interesting idea that could be explored in future work.

-In Alg. 3, suppose we set \alpha = 1. I think the tradeoff between the communication cost and the number of rounds may also be achieved by replacing the doubling strategy (Line 8 of Alg. 3) with \ell = \ell * \beta. When \beta is only slightly larger than 1, the algorithm takes more rounds with the communication cost of each round being small.
The factor 2 in \ell=\ell*2 determines the rate at which we find an estimated value of k. Indeed, we can accelerate or slow down this rate using parameter \beta. However, once we find this estimated value k <= \ell^* < \beta*k, we use \alpha to reduce the communication cost to an \alpha fraction per round. In other words, since \ell^* is always greater than k, instead of selecting \ell^* elements per round, we choose \alpha*\ell^* elements.

-In the experiment section, what \lambda is used for each experiment?
\lambda is always the approximation guarantee for GreeDi and is 1/(36*\sqrt(min(m,k))) as stated in theorem 4.3. Better approximation factors can be easily integrated.

-The bound in Theorem 4.3 may be further improved: the following paper shows a guarantee of (1-1/e)/ (2\sqrt{k}) with k being the cardinality constraint for GreeDI and also shows that GreeDI is tight at 1/\sqrt(k).
Although, the constant in the denominator is smaller, for truly large datasets, k could be much larger than m and 1/(2*\sqrt(k)) could be lower than 1/(36*\sqrt(min(m,k))) provided in theorem 4.3. Furthermore, 1/\sqrt(k) is tight only if m >= k.

Reviewer 3
-One concern is the additional value of the general submodular cover over the cardinality constrained submodular cover. A discussion on this issue may be interesting, especially with respect to its impact to practical applications.
In many practical applications, we care about solutions with certain quality. Hence, instead of trying to find the best subset under the cardinality constraint, one may want to find the smallest subset of data points such that its utility is guaranteed to reach a desirable value.

Reviewer 4
-Can we see some experimental comparisons of some type to non-distributed solutions?
The centralized greedy solution (Greedy) is provided in all experiments of Figure 2.

Reviewer 5
-The paper gives the example of a log-det probability used with DPP and so on, and claims that this is monotone, but I fail to see why. Also, I don't think the reference given on this talks about log-det at all.
Although f(A) =log(det (K_A)) for a general positive definite matrix K might not be monotone, g(A)=log(det(I+\beta *K)) for a \beta>0, which is used in many ML-applications (and section 2 of [24]), is monotone submodular. Note that in [24], the objective function is log-submodular.

-How is "number of rounds" actually defined? How is it defined for the non-distributed variant?
For the distributed approach, 'r' defined in line 3 of Algorithm 3 is the number of rounds. For the centralized algorithm, the number of rounds is 1.